# Design of a Research Laboratory Drive System for a Synchronous Reluctance Motor for Vector Control and Performance Analysis

**Hamidreza Heidari** [1,*], **Anton Rassõlkin** [1,2], **Ants Kallaste** [1], **Toomas Vaimann** [1,2], **Ekaterina Andriushchenko** [1] and **Anouar Belahcen** [1,3]

1   Department of Electrical Power Engineering and Mechatronics, Tallinn University of Technology, Ehitajate tee 5, 19086 Tallinn, Estonia; anton.rassolkin@taltech.ee (A.R.); ants.kallaste@taltech.ee (A.K.); toomas.vaimann@taltech.ee (T.V.); ekandr@taltech.ee (E.A.); anouar.belahcen@taltech.ee (A.B.)
2   Faculty of Control Systems and Robotics, ITMO University, 197101 Saint Petersburg, Russia
3   Department of Electrical Engineering and Automation, Aalto University, 02150 Espoo, Finland
*   Correspondence: haheid@taltech.ee; Tel.: +37-2561-39797

**Abstract:** Motor-drive systems have the most significant share in industrial energy consumption, which requires a deep study in every aspect of the field. This paper presents a synchronous reluctance motor (SynRM) drive system based on Plecs RT box 1. The system's design provides the opportunity for the open-loop and closed-loop control of the motor and a characteristic performance analysis of the motor. This paper focuses on the hardware implementation of a research laboratory setup and the precise vector control of the SynRM in real-time. The application of the digital controller and inverter to drive SynRM is examined. The voltage, current, and speed transducers were employed for monitoring the protective measures and to control the motor in the closed-loop. The design of the signal conditioning and the intermediary cards for isolation and data acquisition are described in detail. An algorithm is proposed to measure the whole system parameters, including motor, inverter, and cables. Thanks to the RT box 1, the principle of real-time simulation of control algorithms is investigated, and the rapid control prototyping of field-oriented control (FOC) of SynRM was implemented. The simulation of the system was carried out in the Plecs platform, and the results are presented. The experimental results of the implemented control algorithms validate the setup's performance and the control algorithm. Finally, as a study of the motor's performance, the efficiency map of the motor is drawn in different speed and torque ranges.

**Keywords:** motor-drive system; synchronous reluctance motor; parameter identification; rapid control prototyping; field-oriented control

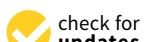



## 1. Introduction

Nowadays, variable speed drives (VSDs) have been applied in many industries for the efficient, high-performance, robust, and precise control of electrical motors [1,2]. VSDs add up to the price of the motor-drive system. Meanwhile, to reach high efficiency of the systems, the application of VSDs is inveatibale [3,4]. Besides, the differences between the prices of the drive systems in the same rated power can be compensated by the lower energy cost. As electric motors consume a significant share of energy in the industry, a lower energy consumption is needed to address the energy crisis in the industry [5]. Besides, the influence of synchronous machines on the distribution of grid stability is a factor that requires more detail in relation to these systems [6]. Therefore, system analysis methods were studied in this study, which aims to develop a more sustainable motor-drive system.

Recent studies, innovations, and the need for a higher efficiency and a better performance support the prospect of the widespread development of motor-drive systems in the industry in the future [7]. These parameters require more study in different areas that can be categorized as follows:

- Optimal design of motors;
- Optimal design of converters;
- Optimal design of controllers.

In recent studies in the literature, researchers have conducted numerous studies to increase the efficiency of motors [8]. Artificial intelligence has offered many popular optimization methods for motor designs. To name a few, genetic algorithms, neural networks, and Taguchi methods have been studied in [9–11]. Additive manufacturing motors are one of the novel developing technologies in this field [12]. Among different motor structures, SynRM represents a superior technology that has lately met the requirements of the IE5 standard [13]. This motor has proved to be a robust structure with a somewhat comparable performance to a well-developed permanent magnet synchronous motor and is potentially as cheap and affordable as an induction motor that lacks the rare-earth magnets in the rotor [8,14].

The control unit of the electric drive system plays a crucial role in system performance. A control system technology with a high performance ensures an efficient and reliable energy conversion. In general, a control unit contains the following:

- Output modules, i.e., a pulse-width modulation (PWM) module which continuously provides switching sequences;
- Peripheral modules for programming and data exchange;
- Main processor for executing and processing control algorithms.

The control algorithms characterize the behavior of the electric drive systems. Hence, to provide a high-performance drive system for motors, a proper control strategy should first be devised [15,16]. The usage of frequency converters controlled by microprocessors provides researchers with the potential to use flexible methods to achieve a better performance. On the other hand, VSDs inject some high-frequency currents into the system [17]. Consequently, some unexpected losses occur, requiring a more precise control algorithm on the drive [3].

There are numerous studies in the literature where the motor-drive systems' simulation or implementation have been discussed. In [18], the authors studied the interaction of the numerical analysis in Magnet with Matlab, which provides an interaction of Magnet and Plecs. This study focused on the simulation of the motor-drive system, where our research work is based on the experimental setup. Besides, our study focuses on the precise implementation of the vector control based on Plecs where the parameters of the whole motor-drive system are measured using the controller and the inverter in the experiments. Moreover, our paper studies the performance of the motor in a wide range of speeds and torques. Regarding the performance analysis of the motor, the authors of [19] have proposed a methodology to study the performance of the motor in the operation phase with proposer control implementation. Our study offers a comprehensive study of the vector control and the performance analysis of the motor.

This paper deals with the design of a research laboratory test bench for the control purpose of SynRM. Firstly, the hardware and the design of the devices in line with the safety tests for the validation of the performance of each device is described separately. Section 2 describes the simulation of the field-oriented control (FOC) of SynRM, which provides information on the modeling of the motor, the inverter and the control system. Then, the interconnection of the devices is investigated. This section describes the rapid control prototyping of the vector control of the SynRM for the sake of real-time simulation. Then a parameter identification algorithm is proposed, which will be employed in the experimental implementation of the control algorithm. The real-time implementation of the control and the results are demonstrated in Section 3. This section employs the implemented setup to analyze the performance characteristics of the motor. Section 5 discusses the results and the outcomes of the study. Finally, a comprehensive conclusion of the study is presented in Section 5.

## 2. Materials and Methods

### *2.1. The Hardware Implementation*

Motor control systems are known as sandboxes of various engineering disciples. This is because many engineering concepts should be applied for an efficient and high-performance drive of the motor. To name a few: power electronics, analog devices, data, digital devices and data, software design, digital signal processing, digital filters, dynamics, mechanics, thermal mechanics, and heat transfer are the main disciplines that should be considered in every motor control setup [20,21]. The complexity of these systems requires a comprehensive scheme for the setup design, where every device is examined accurately in isolated systems and applied in the integrated system along with all other devices. Thus, the setup is divided into three parts of the control unit, the inverter, and the motor. In this section, the design of each piece of hardware and the proper connection is described. Then, an appropriate test is designed for each device to validate the performance with safety consideration, and their proper application in the setup is defined.

### 2.1.1. The Control Unit

The control unit is the main core of the drive system, which simply runs the motor by applying appropriate signals to the gate drivers of the switches in the inverter. In the closed-loop control systems, the control unit collects the feedback from the motor and, based on the current status of the motor, executes the control processes and implements the control algorithms to generate the signal for the application of the command voltages through the inverter. In this paper, the control unit is categorized into three classifications as follows.

The processor

In the control unit, the processor is the intelligent part where all the other devices are employed to send/receive data to/from other devices in the whole drive system. The algorithms were implemented in the processor, which executed the control process. The PLECS RT box 1 "Plexim GmbH, Zurich Switzerland" was utilized in this setup, which benefitted from the Xilinx Zynq processor of Z-7030. The CPU core of this processor is $2 \times 1$ GHz, which denotes that it has two cores, each of which has a clock that beats 1 billion times per second. Regarding the resolution of 16 bits, the RT box 1 can manipulate 16 numbers of the bits in each clock bit. Therefore, in each clock bit, the processor provides a possibility of one billion manipulations for each core in all 16 bits. Furthermore, this device provides 2 mega samples per second, which denotes a high sample update rate. For most motor control applications, a sampling rate of 10 kHz is adequate. Therefore, this device offers a strong processor that perfectly meets the motor control applications' requirements.

The RT box 1 provided the opportunity for rapid control prototyping where the control algorithm was designed in the real-time simulation Plecs platform, and the device was employed as a controller. Furthermore, the high sampling rate and high-fidelity PWM signals, along with the FPGA embedded CPU of RT box 1 offered a versatile processing unit that facilitated the research into control systems for any type of algorithm implementation. Furthermore, thanks to the Xilinx Zynq system-on-chip that benefits from FPGA and CPU cores, the RT box 1 provided a low data transmission latency. In this way, one of the cores was used for communication with the user, and the other core was employed for real-time simulation.

The digital inputs and outputs

The device offered 32 digital inputs and outputs with the logic of 3.3 V and 5 V. The digital breakout board made all digital inputs and outputs accessible via terminal blocks and pin headers. Figure 1 shows a capture of the PWM signals designed with a 10 kHz carrier frequency for two sinusoidal signals. In addition, the signals were designed with 1 μs dead-time, meaning that for each fall and rise of the first signals, the inverted signal had a delay of 1 μs for rising or falling, respectively. This concept secured the operation of the IGBTs in each leg of the inverter. Therefore, with this concept, there were no longer two

turned-on switches at the same time in one leg, and any short circuit in the inverter's legs was avoided.

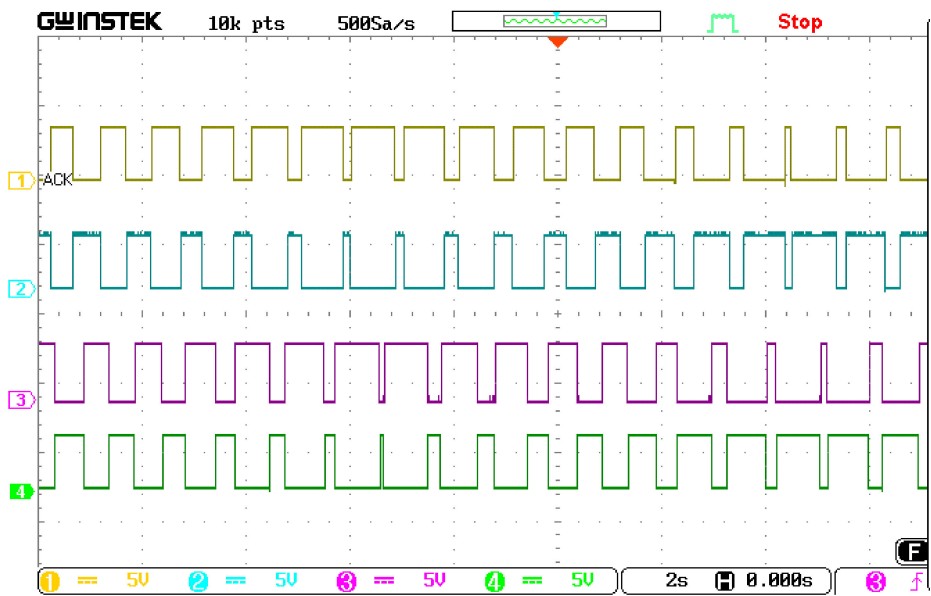

**Figure 1.** PWM outputs for two sinusoidal signals with 1 Hz frequency and a 120 degree shift with 1 μs dead-time: (Cha-1, 3) a and b signals, (Cha-2, 3) inverted correspondent of a and b.

The level shifter

The RT box 1 generated the PWM signals for the inverter with the logic of 3.3 V or 5 V, while the inverter (which will be discussed further) in this setup required 15 V of logic for the gate drivers. This being said, a level translator was required to amplify the PWM signals' level to the gate driver's level. Hence, a board was designed for dual low-side MOSFET drivers. In this board, the MOSFET driver MIC4127YME "Microchip Technology Inc. West Chandler Blvd, Chandler, Arizona, USA" was utilized. The MIC4127YME translated TTL or CMOS input logic levels to output voltage levels that swung within 25 mV of the positive supply or ground, whereas comparable bipolar devices are capable of swinging only to within 1 V of the supply. Therefore, a desirable output of the board was achieved with this board to have the maximum accuracy of the PWM signals. The PCB design was carried out in ALTIUM, the board was printed in double layers, and the components were assembled in the lab. In this design, the IC-Dual MOSFET Driver required ±15 V for operation. To supply the driver, a positive voltage regulator of L7805CV was supplied by a 2.5 mm power jack that was connected to the grid by a low voltage power supply adaptor with the voltage level of 9 V. The regulated positive voltage output of the voltage regulator was fed to a DC-DC converter of MAU209. The DC-DC converter provided an accurate ±15 V for the current driver. The printed board is illustrated in Figures 2 and 3 presents the board's output for a 5 V level PWM signal.

The speed transducer

As the RT box 1 digital inputs work with the TTL logic, all the digital inputs must be compatible with this logic. Hence, to measure the speed and position status of the motor, an incremental encoder of SICK "Myllynkivenkuja 1, Vantaa, Finland" with TTL/RS-422 was employed in the setup. This encoder generates 2500 pulses per revolution to acquire precise information of the motor shaft position and velocity. The device was supplied with 5 V DC to generate a 5 V output of the transducer. This encoder generates 3 pairs of signals. The pairs of A and B were signals with 90 degree shifts that generated 2500 line pairs in each revolution. The sequence of these pairs indicates the direction of the rotation. Thus, one can define the positive or negative direction of the rotation based on the sequence of the pulses. The pairs of Z provided the index pulse that generated one pulse per rotation.

This signal is mainly used in applications where the motor's position is highly critical, and in case of missed pulses for A and B channels, this signal will reset the counter in each revolution. Besides, in some applications, certain positions are of importance. Hence, in these applications (SynRM control), that particular position (the north pole of phase A in SynRM control) was aligned with the index pulse. In this way, regardless of the initial position of the encoder (as well as SynRM), the encoder would reset by the first index pulse, and afterward, the zero position of the encoder output would be aligned with the defined position.

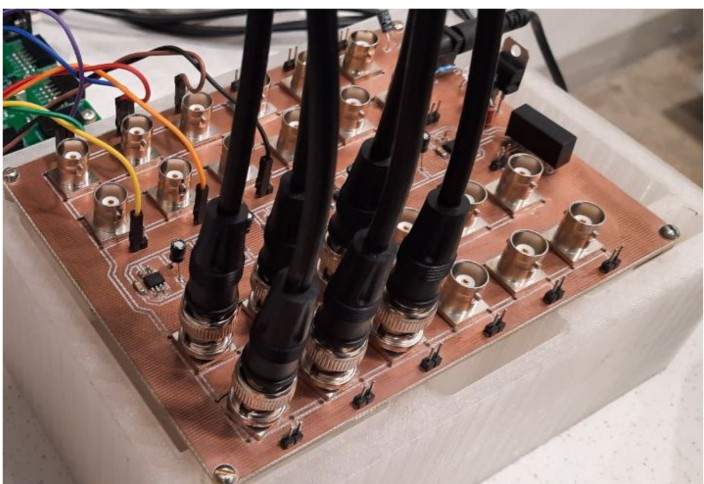

**Figure 2.** The two-layered level shifter board.

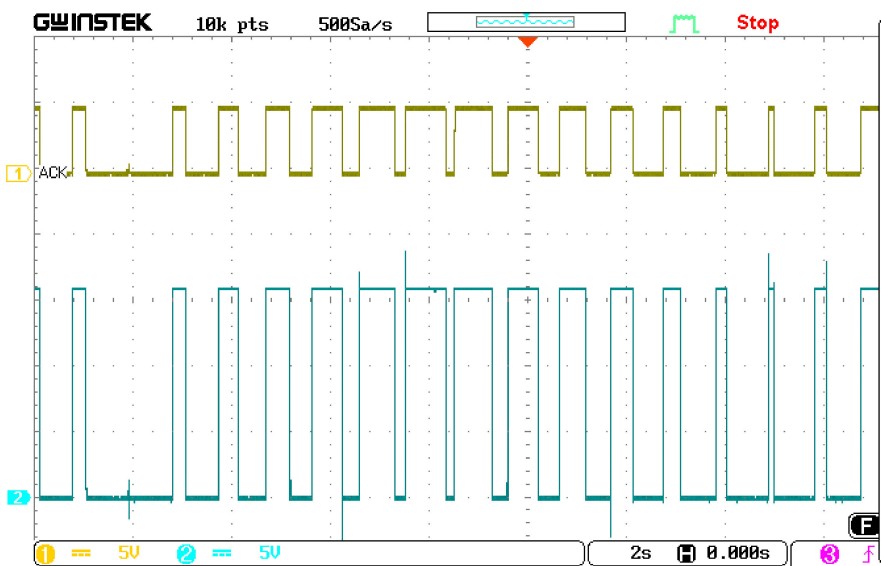

**Figure 3.** (Cha-1) The PWM digital output, and (Cha-2) the translated signal by the level shifter board.

To mount the encoder on the motor, a 3D printed coupling was designed in the SOLIDWORKS "2020, Wyman Street, Waltham, MA, USA" platform and a frame was used to hold the encoder on the motor. The coupling and the holder were printed in the lab. Then, the encoder was coupled to the motor by the coupling and held by the holder, as shown in Figure 4.

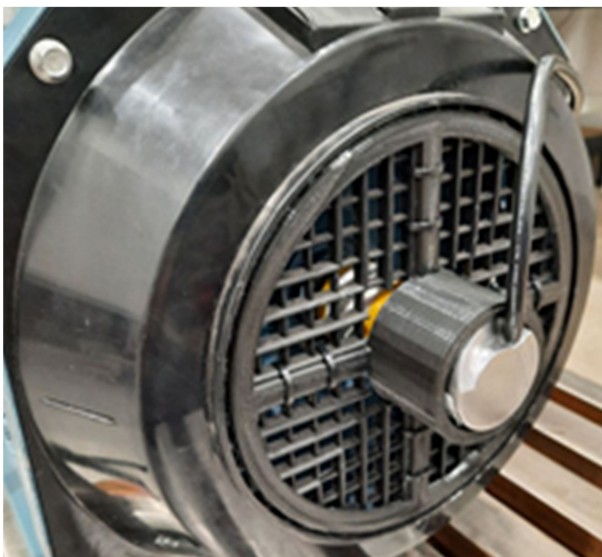

**Figure 4.** The 3D printed holder and coupling for mounting the encoder on the motor.

To test the encoder, it was first supplied by 5 V DC and the outputs of A, B, and an oscilloscope profiled z. Then, the motor was run at a certain speed, and the outputs were displayed. The A and B outputs had the same pulse width and frequency with a 90 degree shift.

To employ the encoder in the real-time setup, the output of the encoder was properly wired to the pin header connection and connected to the RT Box 1 through the digital breakout board. The QEP block of the Plecs simulation setup was employed to count the input pulses of the encoder and to measure the rotor position and speed. The counter of the QEP block was set to (2500 × 4)-1 since all the rising and falling edges were set to count. Then the counted pulses were divided by 360/10,000 to calculate the shaft's position in the dimension of degrees. To convert this value to the radian, it was multiplied by $2\pi/360$. To validate the results, the motor shaft was rotated for one revolution. In this test, the output of the position calculation block counted up to $2\pi$ and dropped to zero. To validate the outputs, a more sophisticated test was completed by rotating the motor with the loading motor at a certain speed. Later, the speed and position information were employed in the closed-loop control of the motor, and the results are displayed in the next section.

The analog inputs and outputs

The RT box 1 offered 16 analog input and output channels with −10–10 V and −5–5 V voltage ranges. The analog input impedance was adjustable to 1 MOhm and 1 ohm with a capacity of 24 pF. The Analogue Breakout Board made the analog inputs and outputs individually accessible via BNC sockets. The breakout board provided the opportunity of reading the single-ended or differential inputs. In this study, the channels were set to −10 to 10 V and 1 kOhm impedance of inputs read in single-ended mode.

The analog transducers

For data exchange, the intermediary devices were designed and exploited in the setup to obtain feedback from the power unit and provide reliable information about the system's status in terms of voltage and current. Hence, the transducers offered a safe connection between the control unit and the system's power unit. The applied transducers in this setup were as follows.

The current transducers

For most of the drive applications, the high accuracy of the current feedback was of paramount importance. In this setup, the hall effect LEM "LEM, Bern, Switzerland" current sensors were applied to obtain the current feedback from the motor. As the working point of the system reached up to 30 A, the current transducer of LA 55-P was opted for, which provided the opportunity of measuring up to ±70 A of AC with a supply of ±15 V while keeping the accuracy of ±0.65%. A two-layer board was designed in Altium to assemble

on a printed circuit board. In this design, the current transducer required a ±15 V supply for operation. For this purpose, a positive voltage regulator of L7805CV was supplied by a 2.5 mm power jack connected to the grid by a low voltage power supply adaptor with the voltage level of 9 V. The regulated positive voltage output of the voltage regulator was fed to a DC-DC Converter of MAU209. The DC-DC converter provided an accurate ±15 V for the current transducer. The current signal output of L.A. 55-P was connected to the ground of the circuit with a 100 Ohm resistance to generate a precise sensor voltage output. BNC outputs obtained this output. The board was printed in the lab, and the components were assembled. The board was tested, and FLUKE current clamps verified the performance of the board in the lab. To provide a handy device for three-phase connections, the board was placed in a box with 3-phase inputs and outputs, and the supply socket and the BNC outputs were also accessible. Figure 5 shows the designed current transducers board and the Fluke current clamp to validate the outputs.

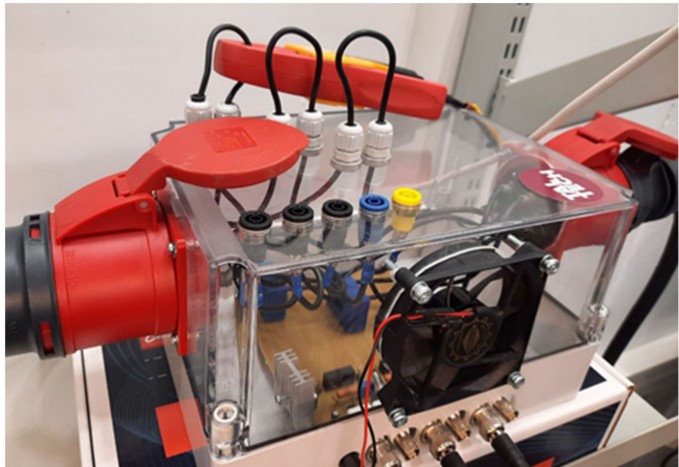

**Figure 5.** The current LEM current transducers PCB and FLUKE current clamp meter.

To test the current transducers, a three-phase voltage input was applied to a three-phase resistance. Figure 6 shows the setup for the validation of the current transducers box. The oscilloscope profiled the three-phase currents of the lines. The outputs of the transducers are illustrated in Figure 7. The sensors correctly measured the three-phase currents, where they showed the same magnitude and frequency with a 120 degree phase shift. To validate the results, the Fluke current clamp was placed in phase a. It can be seen that the current in phase a, which was measured by Fluke, was nearly identical to the current that LEM measured. However, the LEM sensor had a higher accuracy and a bigger bandwidth (200 kHz). Hence, more harmonics were measured by this sensor. One can notice that the scale of the LEM sensors was 10 times bigger than Fluke, which showed that it generated a 10 times bigger output to the same measured current amplitude. It is worth mentioning that using the LEM sensor made the measurement of the DC possible, which was not possible with the Fluke current sensor. Measuring the DC currents is critical for the drive applications. For instance, for the standstill parameter identification test, the measurement of the DC currents is required.

The voltage transducers

The voltage transducer of LTS 25-np "LEM, Bern, Switzerland" was chosen to measure the voltages of the DC link and the line voltages of the motor. The PCB design was designed and prepared in Altium. This board is under work to print and assemble. A differential voltage probe of Pico was utilized in the setup, temporarily.

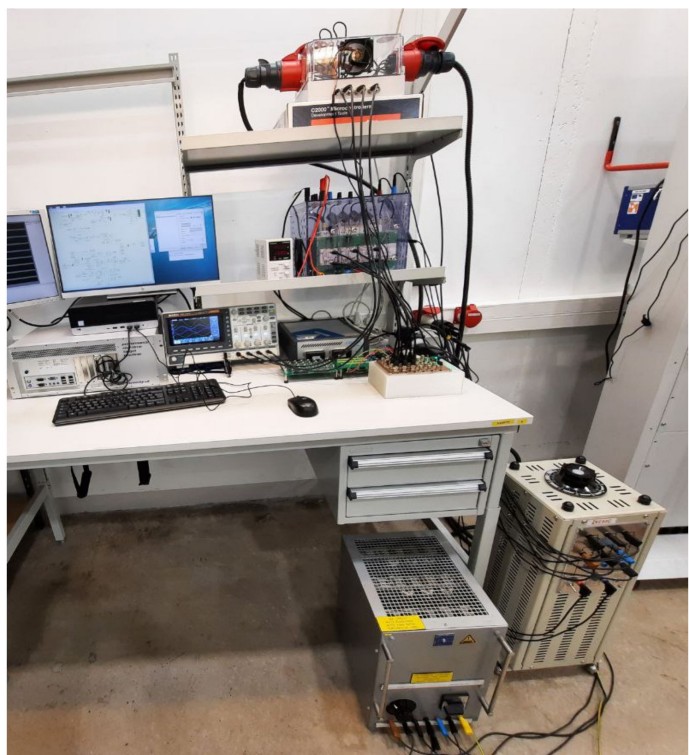

**Figure 6.** The setup for the current transducers box and the loading test of the inverter.

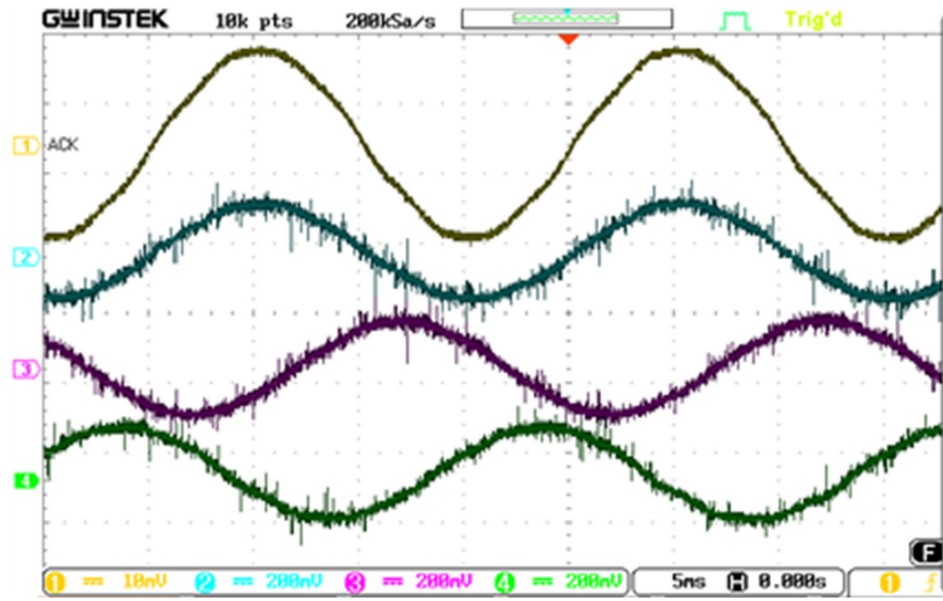

**Figure 7.** The outputs of the current transducers after applying a three-phase voltage to a three-phase resistance, (Cha-1) FLUKE current sensor on the first phase, (Cha-2, 3, 4) LEM current sensors to phase a, b, and c, respectively.

### 2.1.2. Inverter

In this setup, the SEMITACH model of SEMICRON inverters was utilized to produce a desirable set of voltages. To provide a safe operation of the inverter, several tests were designed with this inverter. The inverter worked with the nominal current of 30 A and DC-link of 750 V. The three-phase power inputs and outputs were connected with 4 mm banana connectors. To drive the gate drivers through the level-shifter, 15 V BNC connectors were used. In this inverter, the DC link allowed for the application of a desirable DC voltage.

The snubber capacitors were placed as close to the IGBT module as possible to minimize the inductance between the switches and the capacitor. The reverse bias safe operating area of a switching IGBT is square, which means that there is no need to pull the voltage down to zero before reapplying the current or reverse voltage. This means that the IGBT can be switched at full current and full voltage. As far as the switching characteristics of the IGBT alone are concerned, there is theoretically no need for a snubber unless there is a drastic need to reduce the switching losses. However, a snubber is required to deal with short-circuits and the parasitic inductances of the complete switching loop. External elements forced us to use a snubber, and the snubber will be designed according to the converter's mechanical design. The capacitor snubber was only used for reducing the over vs. the IGBTs turn off by applying zero voltage to the gate drivers. A comprehensive description of the commutation is presented below.

Turn-on: 0 ... t1 (blocked transistor): Gate current will be triggered by applying a control voltage. Up to the charge quantity, the current solely charges the gate capacitance. The gate voltage rises. As the gate voltage is still below the threshold voltage, no gate current will flow during this period.

Turn-on: t1 ... t2 (rise of gate current): As soon as the gate voltage has reached the gate voltage threshold, the transistor is turned on, first passing the active region. Gate current rises to load current level (ideal free-wheeling diode) or even exceeds load current pattern for a real free-wheeling diode. Similarly, gate voltage, which is connected to the collector current in the active region by the transconductance, will increase. Since the free-wheeling diode can block the current only at t2, the collector-emitter voltage will not drop considerably up to t2. At t = t2 the charge gate signal will flow into the gate.

Turn-on: switching interval t2 ... t3 (transistor during turn-on): When the free-wheeling diode is turned off, the collector-emitter voltage will drop almost to the on-state value by time t3. Between t2 and t3, the drain current and gate-source voltage will still be coupled by transconductance; therefore, the collector-emitter voltage remains constant. While the collector-emitter voltage is decreasing, the Miller capacitance is recharged by the gate current with the charge quantity. By t = t3 charge will flow into the gate.

Turn-on: t3 ... t4 (saturation region): At t3 the transistor is turned on and its curve will have passed the pinch-off area to enter the ohmic area. The gate voltage and collector current are no longer coupled. The charge conducted to the gate at this point affects a further increase in gate voltage up to the gate control voltage. Since the collector-emitter on-resistance depends on the collector current and the gate voltage, the on-state voltage may be adjusted to the physical minimum by the total charge quantity conducted to the gate. The higher the collector-emitter voltage (or commutation voltage), the bigger the charge required to reach a certain gate-emitter voltage.

Turn-off: During turn-off, the described processes will run in the reverse direction; the charge has to be conducted out of the gate by the control current.

Two tests were designed to safely analyze the performance of the inverter as follows.

Gate drivers test

In this test, a DC supply was initially used to supply the gate drivers of the inverter. Then, a DC power supply was utilized to supply the DC link with a limited current range. Then, a function generator was applied to generate PWM signals to the input of the gate drivers. Finally, the outputs of the phase voltages were profiled using a scope to validate the setup's performance. This test showed that the gate drivers were working with the 15 V logic of the PWM. To avoid any in-rush current, a resistor was connected in series with the DC link. Figure 8 shows the test bench for this test.

Loading test for the inverter

To assure the performance of the inverter in loading conditions, the outputs of the inverter were connected in a star connection to a three-phase resistor. Then the switching was carried out on the gate drivers, and the voltages were applied to the resistors. The inverter showed a desirable output which validated the performance. Figure 6 shows the setup for this test. In this test, loading was carried out using a set of pulses commanded

by three sinusoidal voltages applied to a three-phase resistance. The test was carried out through the Plecs RT box 1 to generate the PWM pulses. However, a function generator could be sufficient to generate the three-phase pulses. Therefore, in this test, to apply the dead-time and to avoid any risk of damage to the inverter, the RT box 1 was utilized.

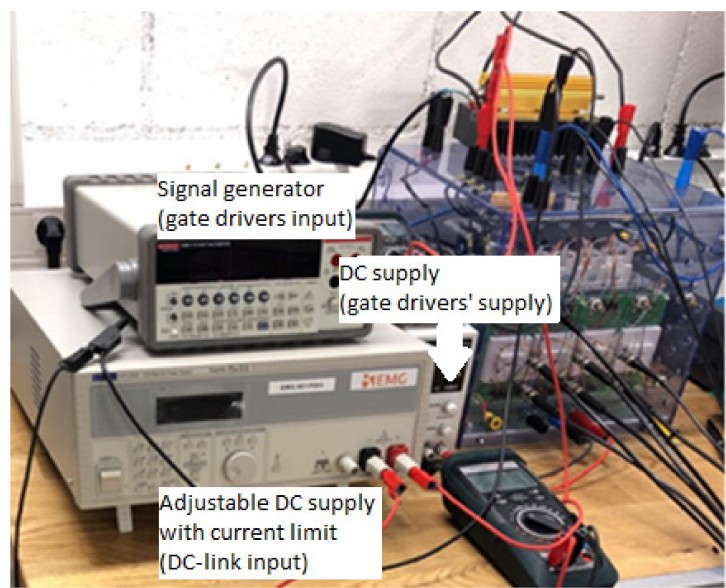

**Figure 8.** The inverters gate drivers test.

Figure 9 shows the results of the inverter loading test. The figure shows the currents of the three-phase resistances sensed by LEM sensors and the phase a current measured by Fluke current clamp. The figure shows that the results were compatible for each phase with a 120 degree shift, and the LEM transducers were capable of measuring higher harmonics and a higher preciseness of the current. Thus, the test validated the performance of all the gate drivers and the power switches, and the inverter could be used in the loop.

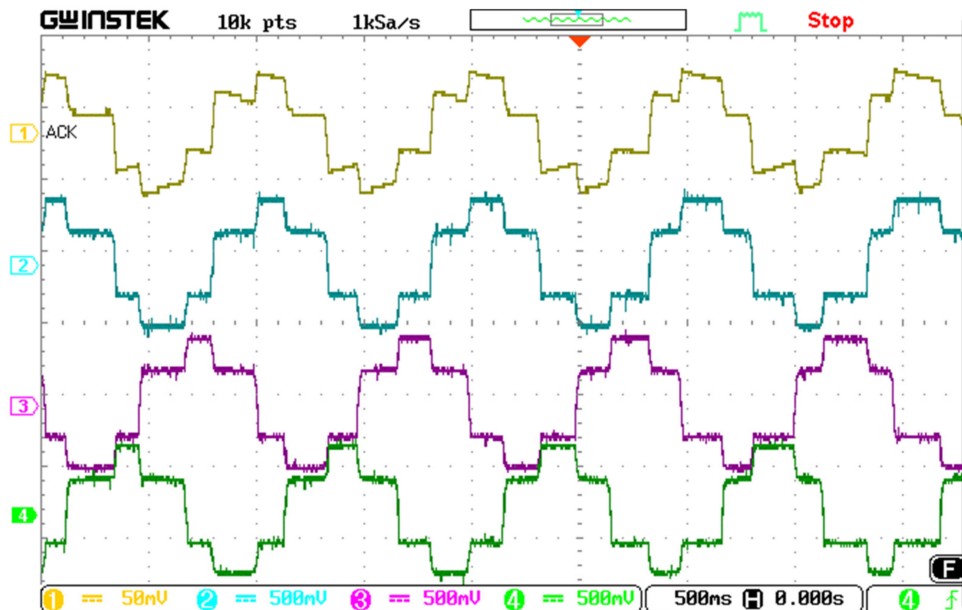

**Figure 9.** The inverter output currents for a three-phase resistance load, (Cha-1) phase a measurements with the Fluke current clamp meter, (Cha-2,3,4) phases a, b, c with LEM current transducers, respectively.

### 2.2. Simulation of FOC of SynRM

In this section, the simulation of the FOC of the SynRM is studied. For the sake of vector control of the motor, the d-q model of the motor was studied in [22]. This model of the motor was simulated in the Plecs platform. The parameters of the motor were measured by the DC decay test and utilized in the simulation. After modeling the motor, the FOC of the motor was implemented in the simulation, as illustrated in Figure 10. For this purpose, the output of the speed and the current inputs of the motor were utilized in the control process as feedback. With the flux position information from the motor's model and the three-phase current feedback, it was feasible to transform the three-phase currents in the stationary reference frame to the synchronous reference frame using Park's transform. By this, the motor was replicated as a DC motor with the correspondence of d-axis current to the flux in the motor and the q-axis current to the motor's torque as the armature and field currents act in the DC motor.

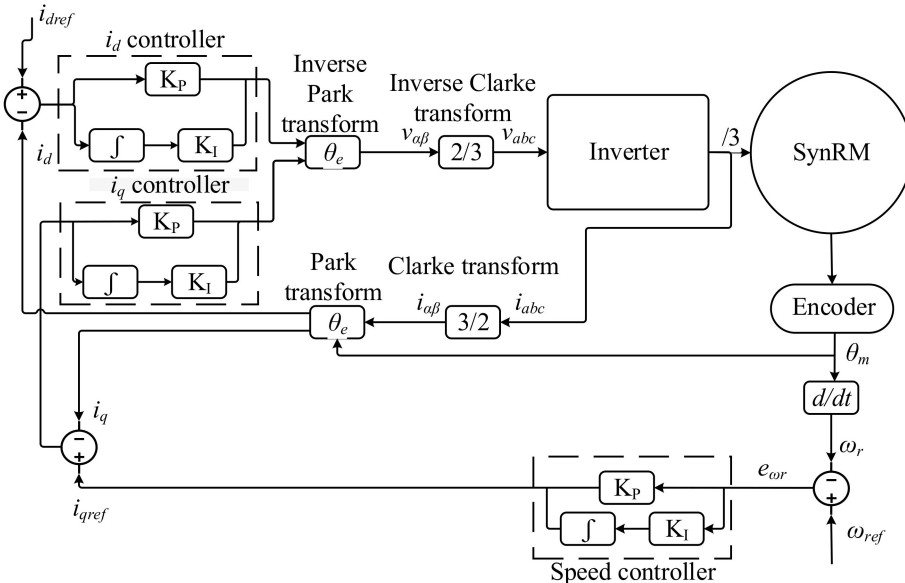

**Figure 10.** FOC of SynRM.

Where, $i_{d,q}$ are the currents in the d and q axes, respectively and $i_{d,q\ ref}$ indicate the reference currents in these axes. $v_{abc}$ represents the phase voltages and $v_{\alpha,\beta}$ and $v_{d,q}$ are the voltages in the stationary and synchronous reference frame, respectively. $\theta$ indicates the position of the motor, which was synchronized with the flux position in this motor. In this simulation, the current reference of the d-axis was a constant value. Using a proportional-integral (PI) controller, the q-axis current was generated by regulating the speed error between the reference speed with the instantaneous speed utilizing a proportional-integral (PI) controller. After generating the errors of the d and q-axes currents by comparing the reference values to the actual values, two PI controllers were employed to regulate the errors and amplify them to generate the d and q-axes' voltages in the synchronous reference frame. Then, the generated voltages were transformed to the three-phase stationary reference frame using the inverse Park transform. Finally, the generated voltages were applied as PWM signals to a model of a three-phase inverter, and the command voltages were applied to the motor's model. In this way, the control was carried out in simulation. Figure 11 shows the results of the simulation of the FOC of SynRM.

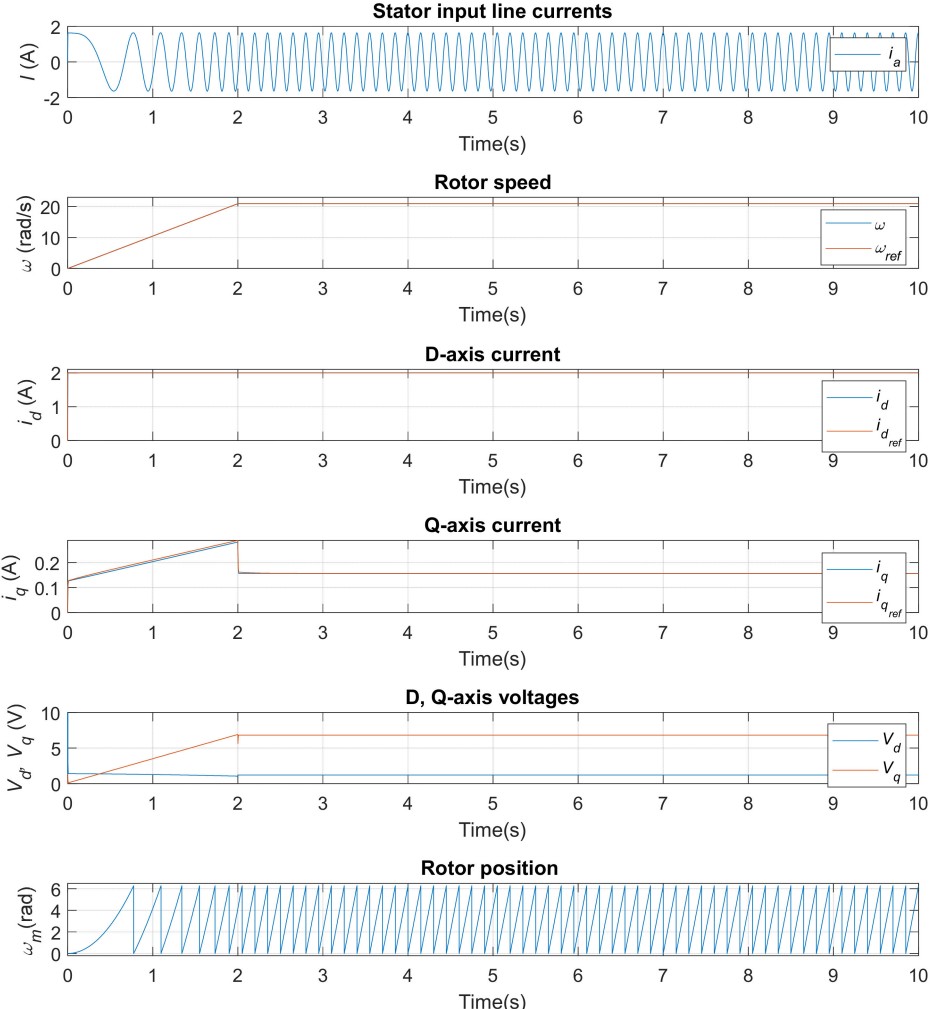

**Figure 11.** Simulation results of the startup ramp response of FOC of SynRM; $I_a$: the measured phase a current, $\omega$: the measures speed ($\omega$) and the reference speed ($\omega_{ref}$), $i_d$: the direct axis current ($i_d$) and the direct axis current reference ($i_{dref}$), $i_q$: the quadrature axis current ($i_q$) and the quadrature axis current reference ($i_{qref}$), $V_{dq}$: the direct and quadrature axis voltages, $\theta_m$: the mechanical angle of the rotor.

### 2.3. The Integration of the Hardware

The proper integration of the devices is a substantial subject that requires a comprehensive understanding of the devices. The connection of the devices, starting from the appropriate connectors to the type of data that each device can send/receive, are critical considerations to take into account. For instance, the secure connectors and wires to connect the inverter to the motor were employed to handle 30 A of a nominal current. For low-voltage analog signals, BNC connectors were utilized. Figure 12 shows the block diagram of the scheme of the setup. The figure illustrates the connection of the power and the signal parts of the system and the integration of all the devices.

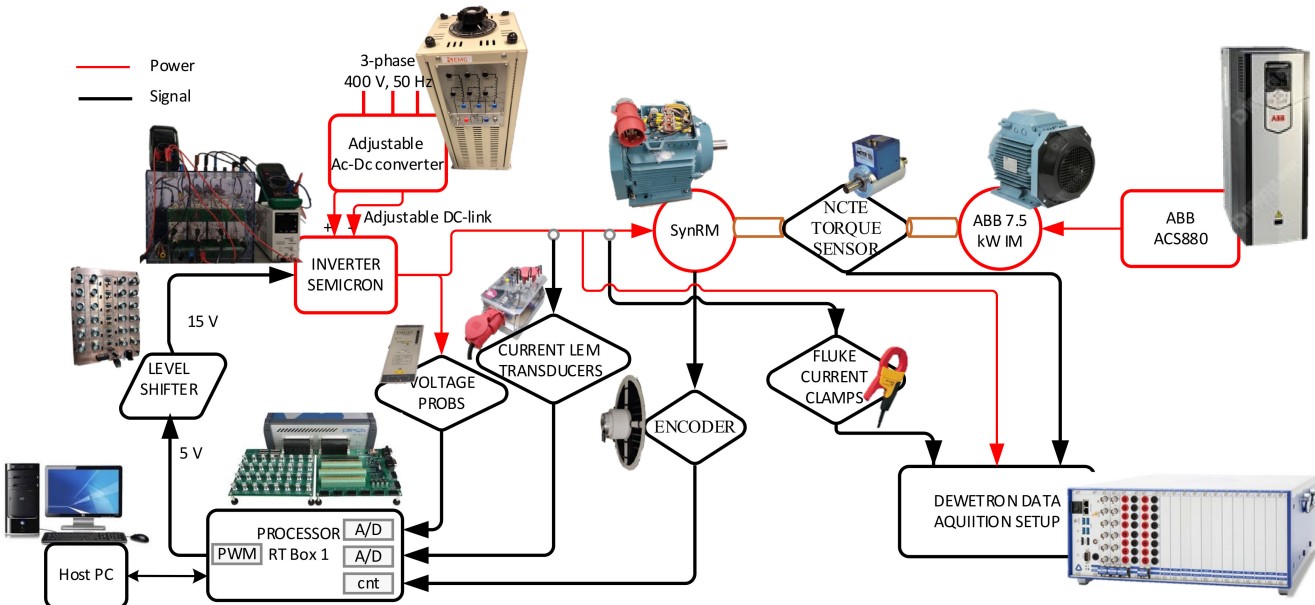

**Figure 12.** The integrated research laboratory setup block diagram.

### 2.4. Rapid Control Prototyping of FOC of SynRM

In this section, the real-time implementation of the FOC of SynRM is discussed. Using the rapid control prototyping of Plecs RT box 1, the closed-loop control of the motor was carried out in real-time, and the vector control of the SynRM was implemented. The FOC of SynRM is described in the next section. First, this cascade control was applied on the motor using an outer controller on the speed with a relatively low sampling frequency because of the low dynamic of the mechanical speed. Then, the inner control was carried out on the currents in a synchronous reference frame with a fairly high sampling frequency due to the high dynamic of the current.

In this method, the motor's current was transformed from the stationary reference frame to the synchronous reference frame using Park's transform. This required precise information of the flux angle, which was obtained by the incremental encoder. Then, an error-driven control concept was applied to compare the reference values of the currents in the d and q-axes. The reference values of the currents in the synchronous reference frame were chosen as constant values for the d-axis, and the q-axis current was generated using a PI controller upon the error value of the commanded and the measured values of speed. Then, the errors of the current were regulated through two PI controllers to generate voltage commands. Using the inverse Park transform, the generated voltages in the synchronous reference frame were transformed to the stationary reference frame. Next, the generated voltages were applied to a PWM module as duty cycles compared with a sawtooth waveform. Finally, the generated signals were applied to the gate drivers of the inverter's switches to apply the commanded voltages to the motor.

### 2.5. Parameter Identification

For the precise vector control of the SynRM, one should consider measuring all the parameters of the motor-drive system. A DC decay test was proposed in our recent study in [22]. In this test, a DC power supply was used to test and measure the motor parameters in an isolated test. This test was initially conducted to simulate the motor's model. However, apart from the measurement approximations and the inaccurate measurement approach of the test, this method only considered the motor parameters' identification, and the inverter and the cables were neglected. To cover this shortcoming, in this study, an algorithm is proposed to calculate the motor parameters and the inverter switches, and the cables in the system. Besides, to tune the encoder information, this method proposes a novel approach to remove the encoder offset using the processor and injecting a

certain set of voltages to the motor. Figure 13 shows the block diagram of the parameter identification method.

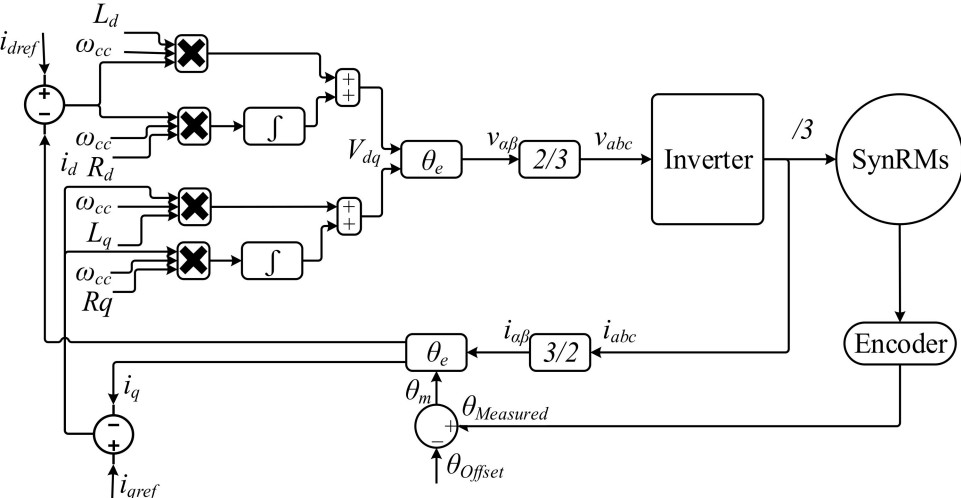

**Figure 13.** Block diagram of the parameter identification algorithm.

Where, $L_{d,q}$, and $R_{d,q}$ are the inductances and resistances in the d, and q axes, respectively. $\omega_{cc}$ is the current regulator bandwidth. In this method, to remove the encoder offset, a non-zero current was commanded to reference the motor in the d-axis. The error between the instantaneous d-axis current and the reference value was regulated using a proportional regulator. In this test, the q-axis current was a zero value. The d and q-axes voltage commands were generated in the synchronous reference frame. These voltages were then transferred to the stationary reference frame using Park transform. Then, the command voltages were applied to the motor using the inverter. As discussed earlier, the d-axis current corresponded to the flux, and the q-axis current corresponded to the torque in the motor. Hence, the applied voltages were anticipated not to generate any torque in the motor unless the flux angle information was incorrect. This could be seen from the q-axis current in the control and a slight rotation in the motor shaft. The test was designed so that a block with a constant value was added to the flux angle information obtained from the encoder. This block was considered as the encoder offset. The current injection was carried out in the d-axis to the motor, and the q-axis current was profiled. If the q-axis current was non-zero, this indicated the wrong flux angle information. The flux angle information was corrected by adding values to the encoder offset block, and the injection of current in the d-axis would not result in current in the q-axis and the shaft movements. It should be noted that the encoder index pulse was considered to indicate the north pole of phase a. In this way, the initial position of the motor was unknown until it reached the position of the index pulse, and the motor position was corrected.

## 3. Results

The experimental setup was integrated, and all the hardware was prepared to implement the closed-loop control in experiments. Figure 14 shows the experimental setup of the designed motor-drive system. In this section, some tests are discussed that were conducted to validate the performance of the real-time control of SynRM. Firstly, a startup ramp will be discussed that aimed to prove the performance of the control in no load. Then the loading condition of the motor will be studied, and then the motor's response to a step reference will be studied at different speeds.

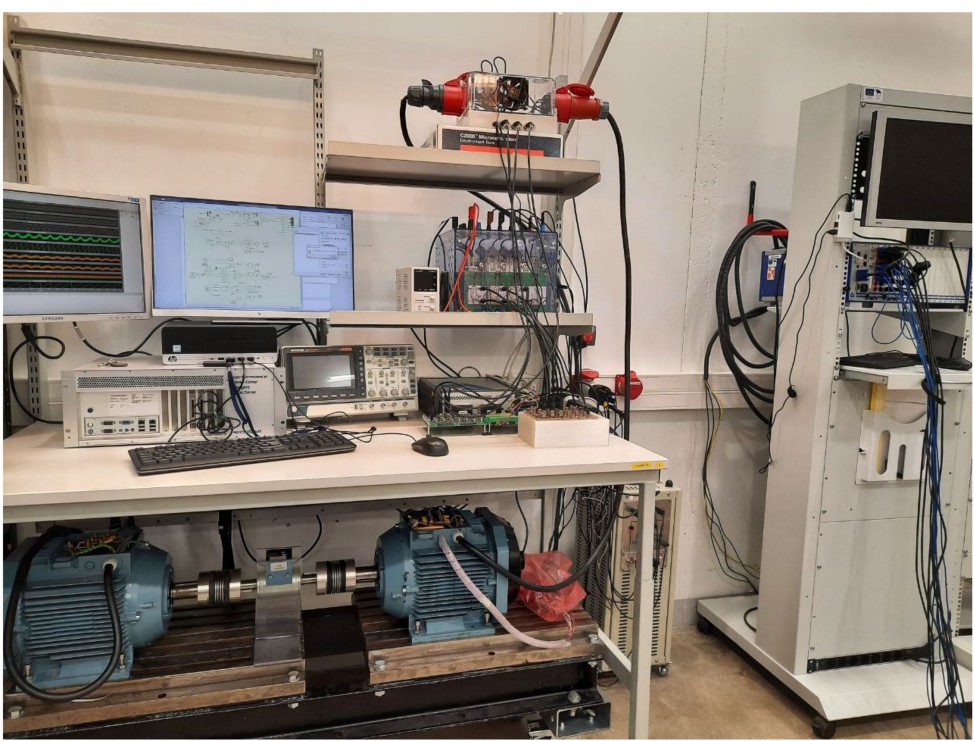

**Figure 14.** The experimental setup of the research laboratory drive system.

### 3.1. Startup Ramp Response

In this test, a ramp speed reference was applied to the motor to analyze the performance of the control. The motor parameters are presented in [22]. In these tests, the reference speed increased up to 200 rpm and then remained constant in that speed. A constant value of 2 A was applied in the d-axis, and the instantaneous value followed the reference value. Figure 15 shows the results of the test. The motor's phase current was sinusoidal, proving the proper control of the motor currents in terms of suppressing the slotting harmonics in the motor compared to the grid-fed control. In the q-axis, the reference was generated by the output of the speed error regulator, which increased to roughly 1.5 A and dropped to nearly zero ampere due to the no-load condition of the motor control. The d and q-axes' voltages were generated and applied to the motor with a slight variation in the steady-state. The motor position information matched with the speed results as well as the current frequency. It is worth mentioning that the experimental results had a slight difference with the simulation due to the approximations in parameter identifications and the motor's modeling. Apart from that, the noises in the environment in the lab resulted in some pulsations, which should be covered in a future study. It should be noted that the frequency of the current matched the speed of the motor, considering the number of poles of the motor. This was true for the mechanical position of the motor's shaft, which was two times bigger than the electrical position of the motor.

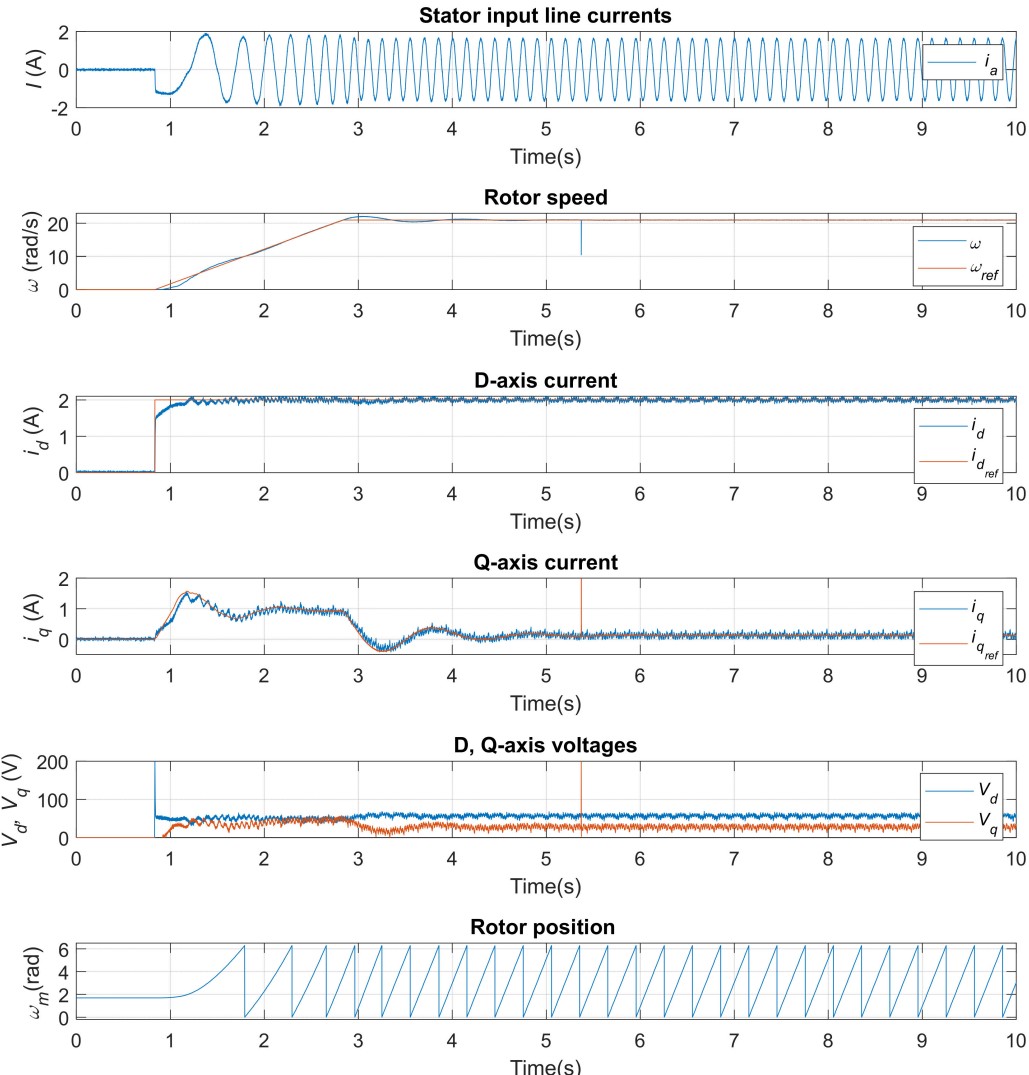

**Figure 15.** Startup ramp response of FOC of SynRM; $I_a$: the measured phase a current, $\omega$: the measures speed ($\omega$) and the reference speed ($\omega_{ref}$), $i_d$: the direct axis current ($i_d$) and the direct axis current reference ($i_{dref}$), $i_q$: the quadrature axis current ($i_q$) and the quadrature axis current reference ($i_{qref}$), $V_{dq}$: the direct and quadrature axis voltages, $\theta_m$: the mechanical angle of the rotor.

### 3.2. Loading Test

The second test that was projected to analyze the performance of the control imposed a step load to the motor and removed the load after reaching the steady-state. For this purpose, the loading motor was coupled to the motor under test. An industrial frequency converter drove the loading motor in the opposite direction. It should be noted that the loading motor was controlled in torque control mode to just inject the reference torque regardless of the output speed. In this way, the loading motor was driven in generator mode since the load and the speed were in opposite directions. By starting the loading motor, a step load of 10% of the nominal torque of the motor was imposed on the motor. To avoid the motor running in the generator mode, a safety algorithm was devised to monitor the DC link of the inverter and the speed and current of the motor. This algorithm was devised to switch off the PWM block in case limits that were defined were passed. After imposing the load with the loading motor, the motor speed was anticipated to drop for a short while and reach the reference value in the steady-state. In the phase of removing the load, the motor speed was projected to have an overshoot for a short while and follow the reference speed in the steady-state. Figure 16 shows the results of the experimental test. The results show that the motor current increased under load to increase the input

power of the motor to dominate the load on the motor shaft. As can be seen, the d-axis current, which corresponded to the flux, remained nearly constant, and the increase in the current was due to an increase in the q-axis current, which corresponded to the torque. As a consequence, the motor generated more torque on the shaft in the opposite direction with the loading motor. Thus, the speed recovered after a small portion of time and followed the reference speed. The voltage in q-axis increased in the loading condition, and the d-axis current dropped negligibly. As can be seen, all the parameters settled down to their initial values after the load was removed.

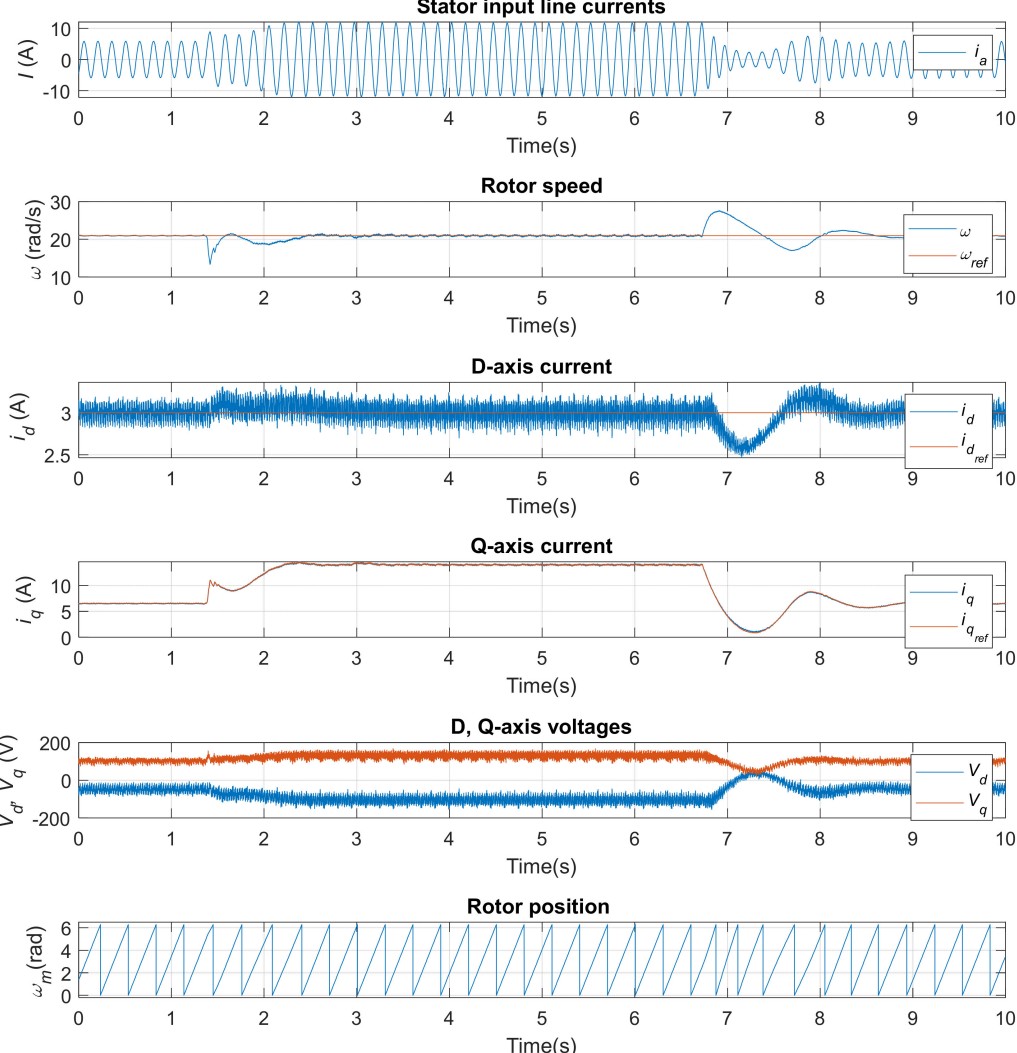

**Figure 16.** Step-load imposing and unloading; $I_a$: the measured phase a current, $\omega$: the measures speed ($\omega$) and the reference speed ($\omega_{ref}$), $i_d$: the direct axis current ($i_d$) and the direct axis current reference ($i_{dref}$), $i_q$: the quadrature axis current ($i_q$) and the quadrature axis current reference ($i_{qref}$), $V_{dq}$: the direct and quadrature axis voltages, $\theta_m$: the mechanical angle of the rotor.

### 3.3. Speed Step Response

To validate the dynamic of the control, a more intensive test was carried out at different speeds of the motor. In this test, step commands were applied to the motor speed reference, and the motor's outputs and inputs, including the speed, were examined. Figure 17 represents the results for the step speed response in different speeds in experiments. The results show that the motor presented an acceptable dynamic, and the motor accelerated to the reference speed in a short time and, with a small overshoot, reached the reference speed without any underdamped settled down on the reference speed, and the steady-state was reached in a short portion of time. This motor dynamic was examined with a step

reference, and the motor showed nearly the same behavior, and the control was carried out. It can be seen from the figure that the motor speeded up to the reference speed in upward increments with an acceptable dynamic as well as the downward step speed responses. The current results show that the controller injected a high current to the motor using the inverter in the step times. This was due to the increment of the speed error because of the sudden increment in the reference and consequently the increment of the differential between instantaneous and reference speed. Hence, as shown in Figure 10, the output of the outer speed controller, which generated the q-axis current reference, saw a sudden increment. Thus, the q-axis current rapidly increased to a high value. Considering the constant d-axis current reference, the increment in the q-axis current resulted in the increment of the current in the motor. The current figure shows that the current amplitude roughly dropped to the initial value in the steady-state due to the same loading condition, which was only the fan and the friction of the motors. The q-axis current and the voltages show that the final values in the steady-state in all the speeds were nearly equal. It is worth mentioning that the voltages saw a dramatic change in the imposing step reference phase, where the step resulted in a very high voltage in both the upward and downward speed change scenarios. It should be noted that the frequency of the current was compatible with the speed and position information of the motor shaft, and the results validate the performance of the control.

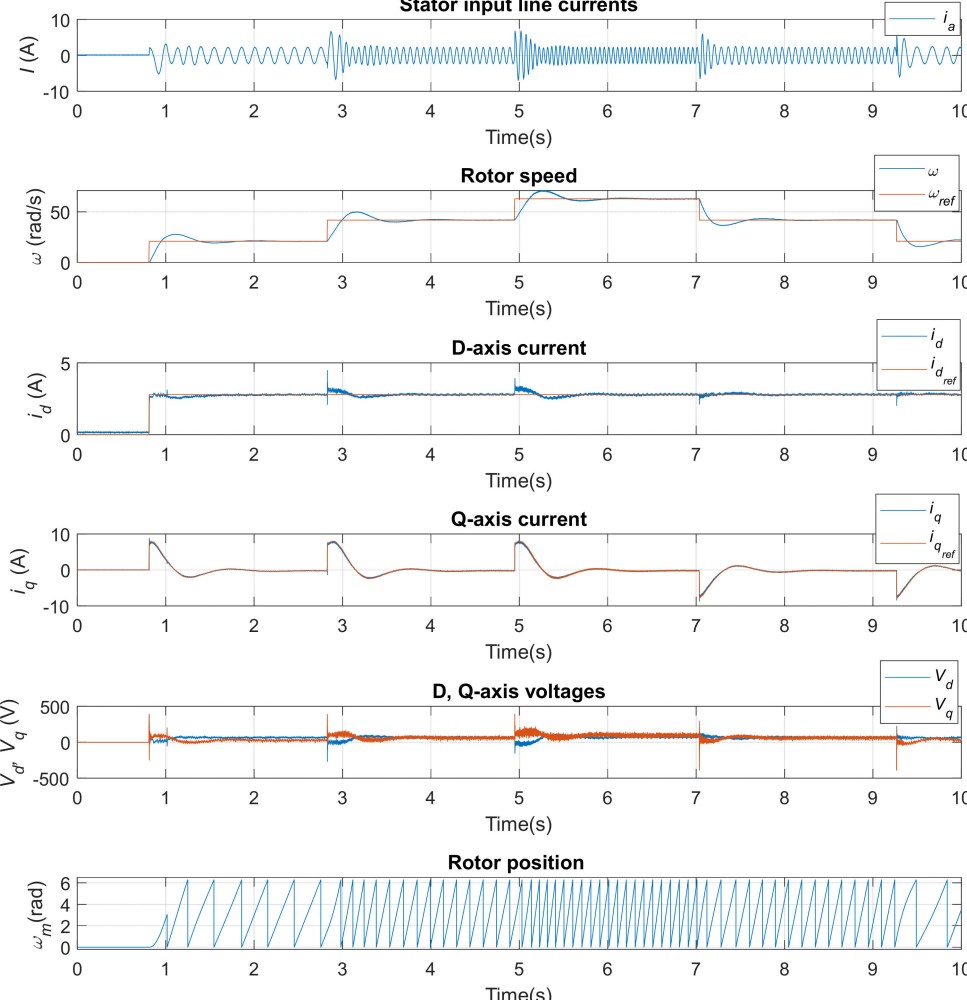

**Figure 17.** Speed step response in different speeds; $I_a$: the measured phase a current, $\omega$: the measures speed ($\omega$) and the reference speed ($\omega_{ref}$), $i_d$: the direct axis current ($i_d$) and the direct axis current reference ($i_{dref}$), $i_q$: the quadrature axis current ($i_q$) and the quadrature axis current reference ($i_{qref}$), $V_{dq}$: the direct and quadrature axis voltages, $\theta_m$: the mechanical angle of the rotor.

### 3.4. Motor Performance Analysis

One of the objectives of this study was to analyze the designed motor's performance characteristics. The motor efficiency at different speeds and under different load conditions was studied. The test was designed to run the motor at different speeds with the step of 200 rpm up to 1800 rpm, which was over the nominal speed of the motor. Then, a ramp load was designed using the industrial frequency converter to ramp up to the nominal torque and impose on the motor. Fluke current clamp meters and Dewetron data acquisition setup measured the motor's input currents and voltages. As well as the inputs, the motor's outputs were sensed using a NCTE torque transducer and registered by the Dewetron data acquisition setup. The power analyzer in Oxygen software was employed to calculate the input power of the motor as well as the output power. Finally, the efficiency of the motor was calculated at different speeds and torques. Figure 18 shows the efficiency map of the motor. The figure shows that the motor presented a big range of high efficiency of around 90% in the nominal speed range and the middle of the nominal torque. A vast region of a nearly high efficiency of 80% was registered in the speeds higher than 500 rpm and torques higher than 15 Nm. This shows that the motor can work in a wide range of speeds and torques with high efficiency. As it was expected for most of the motors, when it came to low speeds, the motor's efficiency dropped to a low value. This was true for the low torques, which required the engineers to opt for the motors regarding the applications in the right power range.

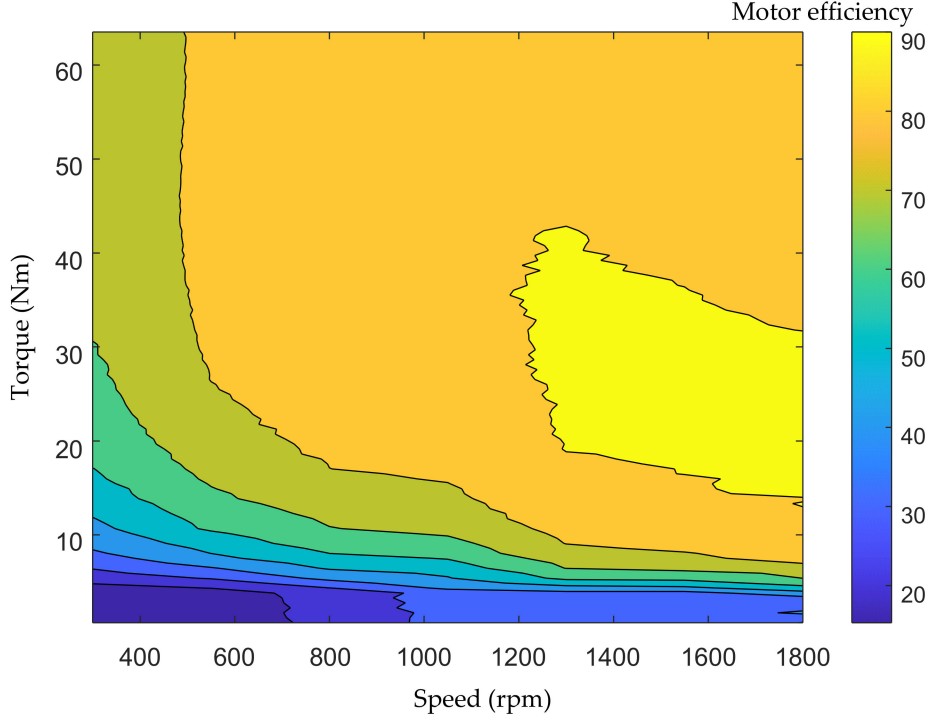

**Figure 18.** Efficiency map of SynRM.

### 4. Discussions

The results of the tests from the controller, inverter, and the inputs and outputs showed that the experimental setup was properly designed and the devices were tuned in the system. The performance of each device was tested in the open-loop with safe tests where the results confirmed them.

The simulations of the motor-drive system were successfully carried out and the results were convincing.

The parameter identification algorithm was properly proposed and tested in experiments. The experimental results validated the performance of the vector control which

certified the preciseness of the measured parameters. The experimental results were in acceptable agreement with the simulation results which validated the correctness of the simulations. The motor was loaded in the experiment that confirmed the high-performance control of the motor with the vector control algorithm. Besides, the dynamic and stability of the control were tested by the step speed reference. The motor showed a desirable dynamic in different speed regions and the transition was smooth where the motor followed the reference speed in a short period.

The performance of the motor was widely tested regarding the efficiency where the motor showed a desirable efficiency with a wide range of high efficiency in the high output power range.

## 5. Conclusions

This paper provides comprehensive guidelines in relation to a case study of the SynRM drive system. The paper dealt with hardware implementation of the device in the setup and integrating all the components. The devices were designed and tested in independent tests. The performance of each device was validated in a safe test, and each device was prepared for implementation in the setup. Moreover, the whole system was integrated to implement the vector control of SynRM. The simulation of the whole motor-drive system was carried out on the Plecs platform. For the sake of precise control, a parameter identification method was proposed to measure the system parameters and obtain the precise angle of the flux using the incremental encoder. The FOC of SynRM was implemented in real-time using rapid control prototyping. The experimental results validated the simulation results, and the performance of the systems was examined in practice. Finally, the efficiency map of the motor was drawn under experimental tests. A future study will involve improving the efficiency of the motor opting for an optimal torque angle.

**Author Contributions:** Conceptualization, methodology, validation, formal analysis, writing–original draft preparation, H.H.; writing—review and editing, resources, funding acquisition, A.R.; project administration, T.V.; investigation, A.K. and E.A.; supervision, A.B. All authors have read and agreed to the published version of the manuscript.

**Funding:** The research has been supported by the Estonian Research Council under grant PSG453 "Digital twin for propulsion drive of autonomous electric vehicle".

**Conflicts of Interest:** The authors declare no conflict of interest.

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
