# Peer review of "Design of a Research Laboratory Drive System for a Synchronous Reluctance Motor for Vector Control and Performance Analysis"

_inventions, doi:10.3390/inventions6040064_

Round 1

Reviewer 1 Report

The manuscript is not suitable for publication in this format. It is not a research work for me, no originality in the proposed method of control, just PI controoler and FOC archeticture with an experimental validations.

Author Response

Thank you so much for your considerations of the manuscript. 

The paper presents a comprehensive study for the real-time implementation of the vector control of the SynRM. the focus was on the experimental setup and the precise control of the motor. For this, all the steps for implementing the hardware were described in detail, and tests to prove the performance of the devices were designed and discussed.

The parameter ID algorithm was proposed to measure the parameters of the system precisely. Moreover, the flux angle information was very precisely obtained. 

we made changes according to the reviewer's comments. 

We hope that the reviewer reconsiders the opinion on the paper.

Best,

Reviewer 2 Report

The contribution of this work is very good and deserves publication. The title has been well formulated and it conveys the focus of the study. Research aims and objectives are well delineated in this contribution.

The authors have a high level of understanding of current research. The contribution is clearly written and the narrative is logical.

The authors used the appropriate techniques for analysis of the research objects in order to meet aims of the study. The accurate interpretation of outcomes, well substantiated by the results of the analysis has been achieved by them. The presentation of the results in terms of the research objectives has been successfully made. Appropriate methods have been used in a well-founded manner. 

The results are interesting.

Nevertheless, I find interesting to discuss the problem  in-rush current more in depth. In fact, you wrote that  a resistor was connected in series with
 the DC link. Please clarify this point and please point out the drawbacks of this possible solution but not the unique one.

Concerning the literature you can consider the following papers related to a similar problem of identification and control to improve the tutorial/Survey character of the paper.

Another interesting question is how to control such a system. In the following three papers you can find three examples of control strategies which can be applied also to the presented system.

Braune et al. “Design and control of an electromagnetic valve actuator” 2006 IEEE Conference on Computer Aided Control System Design, 2006 IEEE International Conference on Control Applications, 2006 IEEE International Symposium on Intelligent Control

P Mercorelli, N Werner “An adaptive resonance regulator design for motion control of intake valves in camless engine systems“,  IEEE Transactions on Industrial Electronics 64 (4), 3413-3422

Mercorelli P., Lehmann K. and Liu S. “Robust Flatness Based Control of an Electromagnetic Linear Actuator Using Adaptive PID Controller”, in Proceedings of the 42nd IEEE Conference on Decision and Control; Maui, HI; United States; 9 December through 12 December 2003.

Mercorelli P.,  “Parameters identification in a permanent magnet three-phase synchronous motor of a city-bus for an intelligent drive assistant”, International Journal of Modelling, Identification and ControlInderscience Publishers, 21 (4), pp. 352-361, 2014.

Y Su et al. “Global finite-time stabilization of planar linear systems with actuator saturation“, IEEE Transactions on Circuits and Systems II: Express Briefs 64 (8), 947-951

Author Response

Thank you so much for the kind comments. We appreciate your considerations of the manuscript. We studied your comments and made changes in the manuscript based on the comments. Please, find the revised version for the changes. Besides, here are the responses to the comments to clarify the questions. 

>>Nevertheless, I find interesting to discuss the problem  in-rush current more in depth. In fact, you wrote that  a resistor was connected in series with
 the DC link. Please clarify this point and please point out the drawbacks of this possible solution but not the unique one.

The series resistor was used to make the test as safe as possible.  This resistor can increase the impedance of the system to avoid inrush current. In the motor-drive system, this resistor can be used to waste the reverse currents as a loss in case of turning to the regenerative mode. 

>>Concerning the literature you can consider the following papers related to a similar problem of identification and control to improve the tutorial/Survey character of the paper.

The references were studied and considered in the text. 

>>Another interesting question is how to control such a system. In the following three papers you can find three examples of control strategies which can be applied also to the presented system.

Thank you so much for sharing these papers. we studied the paper and we have considered considering the papers in the future study.

Reviewer 3 Report

The article deals with electric drives.  The article is interesting and well written. I think, the article should be published after a minor revision (see comments).

Comments:

- Throughout the text, the symbols of the physical quantities should be written in italica font, units in standard font. This is not strictly adhered to. See especially the figures.

- Lines 310, 329, 342, 380, 406, 419, 452, 471, 484, 514 – be careful, there are errors in the links (probably figures or references).

- Figures 7, 9 are difficult to read. They are probably removed from the electronic device without modification.

- 9 self-citations in one article is quite a lot. Authors should consider whether all self-citations are really important.

- References 7, 8, 14, 15, 18 have only the first author listed. There should be all authors stated. The braket is missing at the reference 9 - "(in Press)".

- The following article talks about the influence of rotating machines and on the stability of the distribution grid. I think, the following article should be mentioned in references: T. Petrík, M. Daneček, I. Uhlíř, V. Poulek, M. Libra, „Distribution Grid Stability - Influence of Inertia Moment of Synchronous Machines“. Applied Sciences - Basel, 2020, 10(24), Article ID 9075, DOI: 10.3390/app10249075.

- I recommend acceptance after minor revision.

Author Response

Thank you so much for your kind comments on the manuscript. We appreciate your considerations of the paper.

We made changes on the manuscript based on the comment and we believe the comments improved the quality of the paper. Please, kindly find the revised version for the review of the changes. Hereby, we provide the comments with responses. we hope you find the responses convincing. 

  • Throughout the text, the symbols of the physical quantities should be written in italica font, units in standard font. This is not strictly adhered to. See especially the figures.

We made the changes to the text and the figures according to the comments. 

  • Lines 310, 329, 342, 380, 406, 419, 452, 471, 484, 514 – be careful, there are errors in the links (probably figures or references).

The cross-references were removed by the journal editors. We covered the problem in the revised version

  • Figures 7, 9 are difficult to read. They are probably removed from the electronic device without modification.

As the reviewer mentioned, the figures are obtained from Scope. We tried to magnify the first figure to improve the readability of the figure. for figure 9 we redid the test and more clear results were presented.

  • 9 self-citations in one article is quite a lot. Authors should consider whether all self-citations are really important.

The study is in the defined project in line with the previous research works and the citation is inevitable.

  • References 7, 8, 14, 15, 18 have only the first author listed. There should be all authors stated. The braket is missing at the reference 9 - "(in Press)".

The comments were covered in the revised version. 

  • The following article talks about the influence of rotating machines and on the stability of the distribution grid. I think, the following article should be mentioned in references: T. Petrík, M. Daneček, I. Uhlíř, V. Poulek, M. Libra, „Distribution Grid Stability - Influence of Inertia Moment of Synchronous Machines“. Applied Sciences - Basel, 2020, 10(24), Article ID 9075, DOI: 10.3390/app10249075.

The paper was studied and discussed in the revised version.

Reviewer 4 Report

1- The novelty of this paper is questionable. Please, highlight the contrbution of this paper and compare with the existing literature. What id the difference between this paper and the following:

https://www.mdpi.com/2079-9292/10/17/2154

https://ieeexplore.ieee.org/document/9449505

2- this paper is full of typo. please, revise langauge and figures citation.

3- discuss the snupper and the clamping circuit implemented in this paper.

4- what is the commutation methos used?

5- figure 9 shows strange wavform for the output current, please check and give a clear figure for phase current?

6- Both modes of operation must be considered in both simulation and experimental results ( torque control mode and speed control mode).

7- Flux-weakening mode must also introduced.

8- figure 11 is not clear. please increase font of line  FOR EXAMPLE for V_d and V_q, i see only  V_q

9- FIGURE 18 SHOW EFFICECY MAP. both efficency map of simulation and experimental should be compared.

Author Response

Thank you so much for your comments on the manuscript. We appreciate your considerations of the paper.

We made changes to the manuscript based on the comment and we believe the comments improved the quality of the paper. Please, kindly find the revised version for the review of the changes. Hereby, we provide the comments with responses. we hope you find the responses convincing. 

1- The novelty of this paper is questionable. Please, highlight the contrbution of this paper and compare with the existing literature. What id the difference between this paper and the following:

https://www.mdpi.com/2079-9292/10/17/2154

https://ieeexplore.ieee.org/document/9449505

The first article studies the interaction of the numerical analysis in Magnet with Matlab which provides interaction of Magnet and Plecs. This study focuses on the simulation of the motor drive system, where our research work is based on the experimental setup. our study focuses on the precise implementation of the vector control based on Plecs where the parameters of the whole motor drive system is measured using the controller and the inverter in the experiments. Moreover, our paper studies the performance of the motor in a wide range of speed and torque. This part is studied with the second paper that is recommended. Where our study offers a comprehensive study of the vector control and the performance analysis of the motor. 

2- this paper is full of typo. please, revise langauge and figures citation.

The manuscript was edited with a language expert and the typos and grammar mistakes were removed

3- discuss the snupper and the clamping circuit implemented in this paper.

The snubber capacitors are placed as close to the IGBT module as possible to minimize the inductance between the switches and the capacitor.
The reverse bias safe operating area of a switching IGBT is square, which means that there is no need to pull the voltage down to zero before reapplying the current or reverse voltage. This means that the IGBT can be switched at full current and at full voltage.
As far as switching characteristics of the IGBT alone are concerned, there is theoretically no need for a snubber, unless there is a drastic need for reducing the switching losses. However, a snubber is required for dealing with short-circuit and the parasitic inductances of the complete switching loop. External elements force us to use a snubber, and the snubber will be designed according to the converter mechanical design.
The capacitor snubber is only used for reducing the over voltages.

4- what is the commutation methos used?

The IGBTs turn off by applying zero voltage to the gate drivers. A comprehensive description of the commutation is presented below. 

 Turn-on: 0...t1 (blocked transistor)
Gate current will be triggered by applying a control voltage.
Up to the charge quantity Q
G1 the current iG solely charges the gate capacitance CGE. The gate voltage VGE rises. As VGE is still below the threshold voltage VGE(th), no gate current will flow during this period.
Turn-on: t1...t2 (rise of gate current)
As soon as VGE has reached VGE(th), the transistor is turned on, first passing the active region.
Gate current rises to I
L-level (ideal free-wheeling diode) or even exceeds IL - as indicated in the switching
pattern figure for a real free-wheeling diode.
Similarly, V
GE, which is connected to the collector current in the active region by the transconductance gfc with ID = gfc * VGE, will increase up to the value VGE1 = ID/gfC (time t2).  Since the free-wheeling diode can block the current only at t2, VCE will not drop considerably up to t2.
At t = t
2 charge QG2 has flown into the gate.
Turn-on: switching interval t2...t3 (transistor during turn-on)
When the free-wheeling diode is turned off, VCE will drop almost to on-state value VCE(on) by time t3.
Between t
2 and t3 drain current and gate-source voltage are still coupled by transconductance; therefore, VGE remains constant. While VCE is decreasing, the Miller capacitance CGC is recharged by the gate current iG with the charge quantity (QG3-QG2). By t = t3 charge QG3 has flown into the gate.
Turn-on: t3...t4 (saturation region)
At t3 the transistor is turned on, its curve has passed the pinch-off area to enter the ohmic area. VGE and IC
are no longer coupled by gfs.
The charge conducted to the gate (Q
Gtot-QG3) at this point affects a further increase of VGE up to the gate
control voltage V
GG-. Since the collector-emitter on-resistance RCE(on) depends on IC and VGE, the on-state
voltage V
CE(on) = IC * RCE(on) may be adjusted to the physical minimum by the total charge quantity QGtot
conducted to the gate.
The higher the collector-emitter voltage V
CE (or commutation voltage), the bigger the charge QGtot required to reach a certain gate-emitter voltage, see VGE(QG) curve.
Turn-off
During turn-off the described processes are running in reverse direction; the charge QGtot has to be
conducted out of the gate by the control current.
For approximations to determine the gate charge quantity required for turn-off, the gate charge
characteristic in the V
GE(QG) curve may be used. 

5- figure 9 shows strange wavform for the output current, please check and give a clear figure for phase current?

We repeated the test and more clear results were presented in the revised version.

6- Both modes of operation must be considered in both simulation and experimental results ( torque control mode and speed control mode).

we have prepared a paper as a developed study for this work that considers the torque control mode. If it is fine with the reviewer, we decided to focus on the hardware implementation and the parameter identification of the system in the paper and publish those studies in the new paper. 

7- Flux-weakening mode must also introduced.

As discussed, the field-weakening is also studied in the new manuscript that the authors prefer to focus on the control of the motor. 

8- figure 11 is not clear. please increase font of line  FOR EXAMPLE for V_d and V_q, i see only  V_q

The changes were applied. 

9- FIGURE 18 SHOW EFFICECY MAP. both efficency map of simulation and experimental should be compared.

As mentioned, the manuscript focuses on the experimental implementation of the vector control and performance analysis. Hence, the experimental results were studied and discussed. 

Round 2

Reviewer 4 Report

Thanks for your revised version.

1-I recommended adding the details of your reply to the first and the fourth comment in the previous review inside the paper as it is important for the implemntaion of the proposed study and aslo interseted for readers. The comments are as follows:

- The novelty of this paper is questionable. Please, highlight the contrbution of this paper and compare with the existing literature. What is the difference between this paper and the following:

https://www.mdpi.com/2079-9292/10/17/2154

https://ieeexplore.ieee.org/document/9449505

-what is the commutation methos used?

2- As this paper deals with the hardware implementaion of the drive system,  the details, parameters and configuration of the used snupper circuit should be added inside the paper.

3-Figure 9 shows the inverter output current in case of resistive load, add the output current in case of the motor.

4-there are some typr Use "DC" instead of " D.C.", double check the whole paper.

Author Response

Thank you for your consideration of our study and your efforts to leverage the readability of the paper. we studied the review and implemented the comments in the paper. Here, are the responses to each comment.

>>The novelty of this paper is questionable. Please, highlight the contrbution of this paper and compare with the existing literature. What is the difference between this paper and the following:

https://www.mdpi.com/2079-9292/10/17/2154

https://ieeexplore.ieee.org/document/9449505

The papers were studied and discussed and compared with our study and the focus of our study was mentioned in the manuscript.

-what is the commutation methos used?

The comment was covered in the revised version. 

2- As this paper deals with the hardware implementaion of the drive system,  the details, parameters and configuration of the used snupper circuit should be added inside the paper.

The comment was covered in the new version. 

3-Figure 9 shows the inverter output current in case of resistive load, add the output current in case of the motor.

The current in the case of the motor was studied in the closed-loop control and a comprehensive profile of the system including the current was presented. Please, refer to figure 15.

4-there are some typr Use "DC" instead of " D.C.", double check the whole paper.

The typos were corrected.
